# PM2.5 from automobile exhaust induces apoptosis in male rat germ cells *via* the ROS-UPR$^{mt}$ signaling pathway

**Cao Wang**[1,2]*, **Yingchi Zhao**[1,2], **Bin Liu**[3], **Zhen Luo**[1,2], **Guangxu Zhou**[1,2], **Kaiyi Mao**[1,2]

**1** Department of Pediatric Surgery, Affiliated Hospital of Zunyi Medical University, Zunyi, Guizhou province, China, **2** Guizhou Children's Hospital, Zunyi, Guizhou province, China, **3** Department of Pediatric Surgery, Longgang Maternity and Child Institute of Shantou University Medical College, Shenzhen, Guangdong Province, China

* 332005664@qq.com

## Abstract

### Objective

To explore the underlying mechanism behind the fine particulate matter's (PM2.5)-mediated regulation of reproductive function in male rats, and to determine the role of vitamins in this process.

### Methods

In all, 32 male *SD* rats were randomized to a control cohort (normal saline), a Vit cohort (vitamin C at 100 mg/kg + vitamin E at 50 mg/kg), a PM2.5 cohort (PM2.5 10 mg/kg), and a PM2.5+Vit cohort (PM2.5 exposure + vitamin C at 100 mg/kg + vitamin E at 50 mg/kg), with eight rats in each cohort. After four weeks of exposure, mating experiments were carried out. Thereafter, rats were euthanized, and the testis and epididymis tissues were excised for hematoxylin-eosin (HE) staining and sperm quality analysis. Apoptosis of testis tissues was quantified via a terminal deoxynucleotidyl transferase-mediated dUTP nick end labeling (TUNEL) assay. Moreover, the testicular oxidative stress (OS)-, apoptosis- and mitochondrial unfolded protein response (UPR$^{mt}$)-related essential protein expressions were measured via western blotting (WB).

### Results

After PM2.5 exposure, the sperm count and motility decreased, while sperm abnormality and the apoptosis index increased. HE staining showed that the number of spermatogenic cells decreased. WB showed that the PM2.5 group had decreased expressions of superoxide dismutase (SOD), nuclear factor E2-related factor 2 (Nrf2), and B-cell lymphoma-2 (Bcl-2) ($p < 0.05$), increased expressions of malondialdehyde (MDA), Bcl-2 associated X protein (Bax), and Caspase3 ($p < 0.05$), and downregulated expressions of C/EBP homologous protein (CHOP), heat shock protein 60 (HSP60), and activating transcription factor 5 (ATF5) ($p < 0.05$). These were all reversed by vitamin intervention.

**Data availability statement:** We have uploaded the original article data to the Dryad database. DOI: S1 https://doi.org/10.5061/dryad.4f4qrfjph;S2 https://doi.org/10.5061/dryad.8kprr4xxq; S2 https://doi.org/10.5061/dryad.pzgmsbcz2

**Funding:** This work was supported by the Guizhou Provincial Health Committee support program (grant no. gzwjkj2023–413), and the Zunyi Science and Technology Fund HZ (2023) 238. The funders had no role in study design, data collection and analysis, decision to publish, or preparation of the manuscript.

**Competing interests:** The authors have declared that no competing interests exist.

## Conclusion

PM2.5 from automobile exhaust disrupts male reproductive function. A combination of vitamins may protect reproductive function via the reactive oxygen species (ROS)-UPR$^{mt}$ signaling pathway.

## 1. Introduction

In recent years, male infertility has become a worldwide concern, and its critical pathogenic factor is a decline in sperm quality[1]. As atmospheric pollution increases, the male infertility rate increases yearly. The potential damage of atmospheric pollution to sperm has been reported, but its specific mechanism remains unclear.

Fine particulate matter (PM2.5) from automobile exhaust is considered the primary pollutant in the urban atmosphere in China. Characterized by a complex composition, the greatest danger, and the strongest pathogenicity, PM2.5 carries harmful substances into the body during breathing to damage systemic tissues and organs[2]. The toxicological study of PM2.5 in the human body has focused mainly on the respiratory, cardiovascular, and nervous systems but rarely on the male reproductive system. PM2.5 generates reactive oxygen species (ROS) within cells, stimulating oxidative stress (OS) in the body [3]. Moreover, a large amount of ROS will attack proteins and DNA and damage mitochondria, resulting in cell death[4]. However, the specific molecular mechanism of PM2.5-driven mitochondrial destruction within germ cells remains unclear. The mitochondrial unfolded protein response (UPR$^{mt}$) is a mitochondrial protective response against OS, which can improve both mitochondrial and cell survival rates[5]. UPR$^{mt}$ maintains cell stability by upregulating the expressions of mitochondrial protective genes, such as chaperone proteins [e.g., heat shock protein 60 (HSP60) and HSP90] and proteases [e.g., caseinolytic protease P (ClpP)], restoring misfolded proteins and degrading damaged proteins[6]. Activating transcription factor 5 (ATF5) is responsible for transcriptional upregulation of chaperone proteins and proteases during mitochondrial stress [7]. In addition, ATF5 and C/EBP homologous protein (CHOP) are critical regulators of UPR$^{mt}$ [8]. UPR$^{mt}$ has been studied mainly in terms of cell senescence, cardiovascular disease, neurologic disease, and tumors, but its role in the reproductive system has rarely been reported. Thus, the influence of PM2.5 exposure from automobile exhaust on UPR$^{mt}$ in germ cells is still unclear.

Oxidative stress can lead to functional impairment in sperm production. The combined use of vitamin C and vitamin E, which have a synergistic antioxidant effect, has been shown to mitigate the impact of environmental toxins on the testes[9].In this study, the changes in and relationships among OS, UPR$^{mt}$, and apoptosis in groups exposed to PM2.5 of different concentrations, as well as the changes in reproductive function after combined vitamin intervention, were observed to explore the toxicological mechanism of PM2.5 from automobile exhaust for inducing germ cell damage. This research inspires novel ideas and insights for preventing and treating automobile exhaust-derived PM2.5-induced reproductive impairment.

## 2. Materials and methods

### 2.1 Extraction and treatment of PM2.5

PM2.5 was collected using atmospheric particulate samplers on heavily traveled roads of Zunyi City, Guizhou Province, China, from September 2021 to December 2021. After the end of sampling, the filter membrane loaded with PM2.5 was cut into square blocks of 1 cm$^2$ and soaked in normal saline, followed by ultrasonic vibration four times (40 min), filtration with

sterile medical gauze, and a 30-min centrifugation at 12,000 rpm at 4°C. Then, the suspension was freeze-dried using a freeze dryer, and the resulting solid PM2.5 was maintained at − 80°C. A PM2.5 suspension at the required concentration was prepared within 24 h before use by adding sterile saline and shaking overnight in a constant 37°C shaker at 60 rpm. The PM2.5 suspension prepared could remain at 4°C for one week and was shaken thoroughly before use.

## 2.2 Animals and grouping

Laboratory animals were acquired from the Laboratory Animal Center of Zunyi Medical University [license No. SYXK (Guizhou) 2021-0004]. This study passed the ethical review of laboratory animal welfare of the participating institution (No.: zyfy-an-2023-0013). In total, 32 SPF-grade male SD rats (30 days old, 75–95 g) were maintained in a room with a barrier system and fed independently. They were randomized to four groups: control (transnasal drip of normal saline), Vit (gavage with vitamin C at 100 mg/kg + vitamin E at 50 mg/kg), PM2.5 (transnasal drip of PM2.5 10 mg/kg), and PM2.5 + Vit (gavage with vitamin C at 100 mg/kg + vitamin E at 50 mg/kg, and transnasal drip of PM2.5 10 mg/kg 30 min later), with eight rats in each group. The treatment consisted of five days of exposure followed by two days of rest, lasting four weeks.

## 2.3 Mating exeriments and specimen collections

After a four-week exposure period and a subsequent two-day rest, 32 twelve-week-old female SD rats were paired with the experimental male rats in a 1:1 ratio for one week. Subsequently, Following a 4-week exposure, the rats were weighed, anesthetized, and sacrificed by cervical dislocation. Then, bilateral testis and epididymis were harvested and weighed. The right testis and epididymis underwent fixation in 10% formalin for histopathological analysis and terminal deoxynucleotidyl transferase-mediated dUTP nick end labeling (TUNEL) assay. The left testis was immediately stored in liquid nitrogen at − 180°C for subsequent sperm quality analysis.

## 2.4 Organ coefficient

The left testis and epididymis were weighed for organ coefficient computation as follows: organ coefficient = (organ wet weight [g]/body weight [g]) × 100%.

## 2.5 Sperm quality analysis

We prepared the sperm filtrate by slicing and dissolving the left cauda epididymis in normal saline at 37°C. Then, 10 μL of suspension was aspirated into a blood cell counting plate (25 × 16), and the total, viable, and abnormal sperm counts were measured in 5 fields of view under a light microscope (400×) [10, 11]. The sperm morphology was observed under the microscope (400×), and double-headed, double-body, double-tailed, neck-folded, body-folded, and head-deformed sperm were considered abnormal. Sperm abnormality rate = abnormal sperm count/total sperm count × 100%, and sperm survival rate = viable sperm count/total sperm count × 100%.

## 2.6 Histopathological examination of testis

The right testis and epididymis were excised for histopathological examinations. Specifically, the testis was PBS-rinsed, prior to a 24-h fixation in 10% paraformaldehyde, dissection at the sagittal position, fixation for another 24 h, and embedding in paraffin, followed by sectioning, deparaffinization, hematoxylin-eosin (HE) staining, dehydration, and transparentization.

Finally, the morphologies of the seminiferous tubules and spermatogenic cells, sperm count, and sperm shedding in the lumen were evaluated under a light microscope (Leica, DM3000,Germany).

## 2.7 Ultrastructural changes in the mitochondria of germ cells

After each rat was sacrificed, testis tissue of approximately $2 \times 2 \times 2$ mm was harvested rapidly, fixed in 3% glutaraldehyde, rinsed, re-fixed, rinsed again, dehydrated with gradient acetone, and embedded, followed by ultrathin sectioning (approximately 60–90 nm) using an ultramicrotome (Leica,EM UC6, Germany). After a15-min uranium acetate staining and 2-min lead citrate staining, the sections were photographed under a transmission electron microscope (TEM, JEM-1400Flash, Japan).

## 2.8 Apoptosis assay (TUNEL staining)

Testis tissue apoptosis was determined using an apoptosis assay kit (YF®488, UEIandy, Suzhou, China) and DAPI (ab104139, Abcam) according to the instructions. Under a fluorescence microscope (Leica, DMi8,Germany), normal cells nuclei exhibited blue fluorescence, while apoptotic cell nuclei exhibited green fluorescence. Three replicates were set up for each group. Apoptosis index (%) = (number of apoptotic cells/total number of cells) × 100%.

## 2.9 Western blotting (WB)

From removal from − 180°C, testis tissues were thawed on ice. Then, protein was extracted from 100 mg of tissues and quantified according to the instructions. Equal protein quantities were electrophoresed on sodium dodecyl sulfate polyacrylamide gel electrophoresis (SDS-PAGE), prior to transfer to a polyvinylidene fluoride (PVDF) membrane, followed by a 2-h blocking in 5% bovine serum albumin (BSA) at room temperature (RT), and incubation with primary antibodies,including SOD(ab13498,Abcam),Nrf2(ab92946;Abcam),M-DA(ab27642;Abcam),ATF5(ab133504;Abcam),

Chop(381679;ZenBio),Hsp60(ab133504;Abcam),Bax(ab32503;Abcam),Bcl2(ab196495;Abcam),Caspase3(ab214430;Abcam) at 4°C for 16h. Following three rinses in tris-buffered saline with Tween-20 (TBST), the membrane underwent a 2-h incubation in secondary anti-rabbit IgG(ab97080;Abcam) at RT, with three subsequent washes. The protein bands were visualized using an ECL(ZETA-Life; San Francisco) solution, prior to photography via an automatic chemiluminescence image analysis system. Finally, the gray value was measured using Image J software, and the protein expression quantified using β-actin as endogenous reference.

## 2.10 Statistical analysis

Data provided as means ± standard deviation. We conducted one-way ANOVA in GraphPad Prism 9.0 (San Diego, CA, USA), followed by normality tests for homogeneity of variance and F-tests first and then Tukey's tests for multiple comparisons. $P < 0.05$ was the significance threshold, and statistical charts were generated using the means ± standard deviation.

## 3. Results

### 3.1 Impacts of PM2.5 exposure on the reproductive capacity of rats

After four weeks of exposure, 32 female SD rats (12 weeks old, 270–330 g) were selected and allowed to mate with male rats at 1:1 for one week. The test result was positive when a female rat gave birth (Table 1 and S1 File). The reproductive capacity significantly decreased among the PM2.5 rats but significantly improved among the PM2.5 + Vit rats.

**Table 1. Pregnancy rate of female rats in each group.**

| Group | n | conception | conception rate(%) |
|---|---|---|---|
| Control | 8 | 8 | 100% |
| Vit | 8 | 8 | 100% |
| PM2.5 | 8 | 2 | 25% |
| PM2.5 + Vit | 8 | 5 | 63% |

https://doi.org/10.5061/dryad.4f4qrfjph

### 3.2 Influences of PM2.5 exposure on rat body and organ weights

Body weights were measured post exposure, and the testis and epididymis were harvested and weighed. As summarized in Table 2 and S1 File, both body and organ weights decreased among PM2.5 versus control rats ($p < 0.05$). Moreover, they substantially increased among the PM2.5 + Vit versus PM2.5 rats, showing marked statistical difference ($p < 0.05$). The relative testis and epididymis weights exhibited no obvious difference among cohorts ($p > 0.05$). These data implied that the PM2.5-exposed group experienced significant reductions in body and organ weights.

### 3.3 Sperm quality analysis results

To clarify the influence of PM2.5 exposure on male rats, we examined the sperm quality in the epididymis (Table 3 and S1 File). Relative to control rats, PM2.5 rats showed diminished sperm count and survival rate and an increased sperm abnormality rate. Relative to the PM2.5 rats, the PM2.5 + Vit rats exhibited an increased sperm count and survival rate and a decreased sperm abnormality rate, with marked statistical differences ($p < 0.05$).

### 3.4 PM2.5 exposure induced morphological abnormalities in the testis and epididymis

Control rats exhibited many layers of spermatogenic cells, an abundant number of sperm, a uniform distribution of spermatogenic cells in the testis, and many sperm in the epididymis were observed (Fig 1A&E and S3 File). PM2.5 rats showed a decline in the layers of spermatogenic cells, the spermatogenic cells were loosely and erratically arranged, the seminiferous tubules were poorly developed, with tissue holes and sperm shedding in the lumen, and the number of sperm in the epididymis decreased (Fig 1C&G). In PM2.5 + Vit rats, cells were arranged regularly in the seminiferous tubules, the number of cells increased, less sperm shedding in the lumen was observed, and the epididymal sperm quantity was enhanced (Fig 1D&H).

### 3.5 PM2.5 stimulated OS

To verify whether PM2.5 induces testicular OS, we performed WB to measure the superoxide dismutase (SOD), nuclear factor E2-related factor 2 (Nrf2), and malondialdehyde (MDA) contents in the testis tissue of the rats (Fig 2 and S2 File). The expressions of SOD and Nrf2 decreased among PM2.5 rats but increased among PM2.5 + Vit rats. In contrast, the MDA content was augmented among PM2.5 rats but decreased among PM2.5 + Vit rats, with marked statistical differences ($p < 0.05$).

### 3.6 Impacts of PM2.5 on mitochondrial destruction and UPRmt activation within germ cells

To further elucidate PM2.5 action on the mitochondria in germ cells, we observed the mitochondrial damage under a TEM (Fig 3 and S3 File). In the control and Vit groups,

**Table 2. Body and absolute and relative organ weights of rats post PM2.5 exposure ($\overline{\chi}$±s, n = 8).**

| Group | Body weight(g) | Testie (g) | Epidymis (g) | Organ Index (%) | |
|---|---|---|---|---|---|
| | | | | Testie | Epidymis |
| Control | 323.1 ± 6.53 | 1.53 ± 0.08 | 0.57 ± 0.04 | 0.48 ± 0.03 | 0.17 ± 0.02 |
| Vit | 324.1 ± 4.52 | 1.58 ± 0.05 | 0.57 ± 0.03 | 0.49 ± 0.02 | 0.17 ± 0.01 |
| PM2.5 | 286.4 ± 7.48* | 1.26 ± 0.06* | 0.48 ± 0.02* | 0.46 ± 0.02 | 0.17 ± 0.01 |
| PM2.5 + Vit | 306.0 ± 5.56# | 1.47 ± 0.06# | 0.52 ± 0.03# | 0.49 ± 0.02 | 0.17 ± 0.01 |

Note:

*$p < 0.05$ *vs*. control rats,

#$p < 0.05$ *vs*. PM2.5 rats.

https://doi.org/10.5061/dryad.4f4qrfjph

**Table 3. Alterations in the post-PM2.5 exposure sperm quality ($\overline{\chi}$ ±s).**

| Group | n | Sperm count (× 106/mL) | Sperm Survival Rate (%) | Sperm malformation rate (%) |
|---|---|---|---|---|
| Control | 8 | 55.84 ± 1.824 | 81.1 ± 2.197 | 5.04 ± 0.715 |
| Vit | 8 | 55.39 ± 1.512 | 81.8 ± 1.645 | 4.80 ± 0.52 |
| PM2.5 | 8 | 23.06 ± 3.783* | 41.1 ± 3.4* | 13.53 ± 3.05* |
| PM2.5 + Vit | 8 | 39.79 ± 1.282# | 56.4 ± 3.791# | 7.07 ± 0.43# |

Note: Data shown as means ± standard deviation of eight rats.

*$p < 0.05$ *vs*. control rats, #$p < 0.05$ *vs*. PM2.5 rats.

https://doi.org/10.5061/dryad.4f4qrfjph

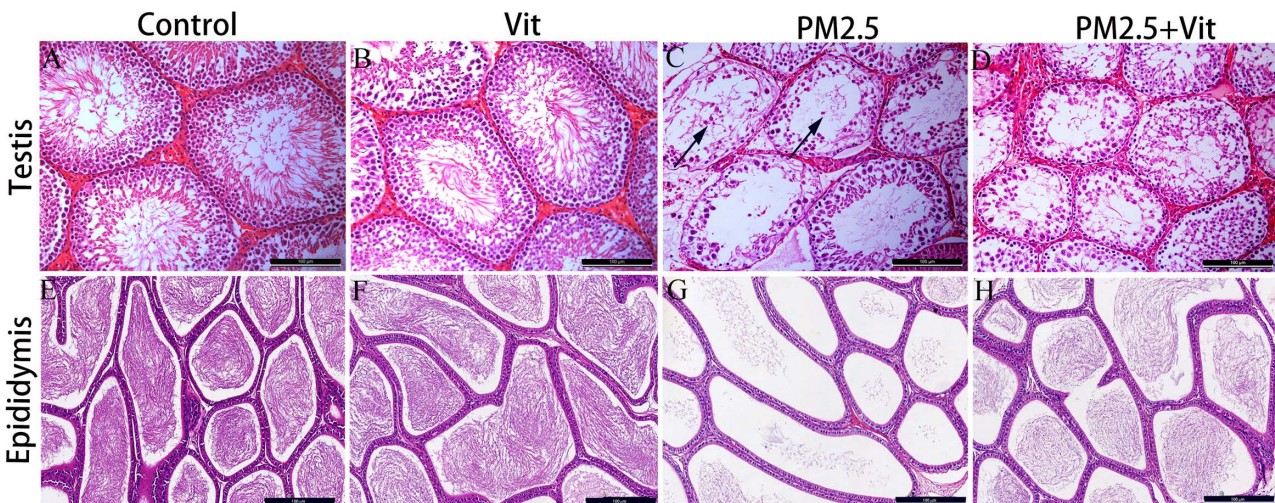

**Fig 1. Morphological changes in the testis and epididymis.** A & E) Control rats revealed many sperm in the seminiferous tubules and a regular arrangement of spermatogenic cells are visible. B & F) There were no significant changes in the Vit versus control rats. C & G) PM2.5 rats exhibited sparse germinal layer in the seminiferous tubules, lumenal vacuolation and sperm shedding (black arrow) were observed, and the sperm quantity was significantly decreased. D & H) PM2.5 + Vit rats showed marked increase in sperm density, which improved relative to the PM2.5 rats. Scale bar = 100 μm. https://doi.org/10.5061/dryad.pzgmsbcz2.

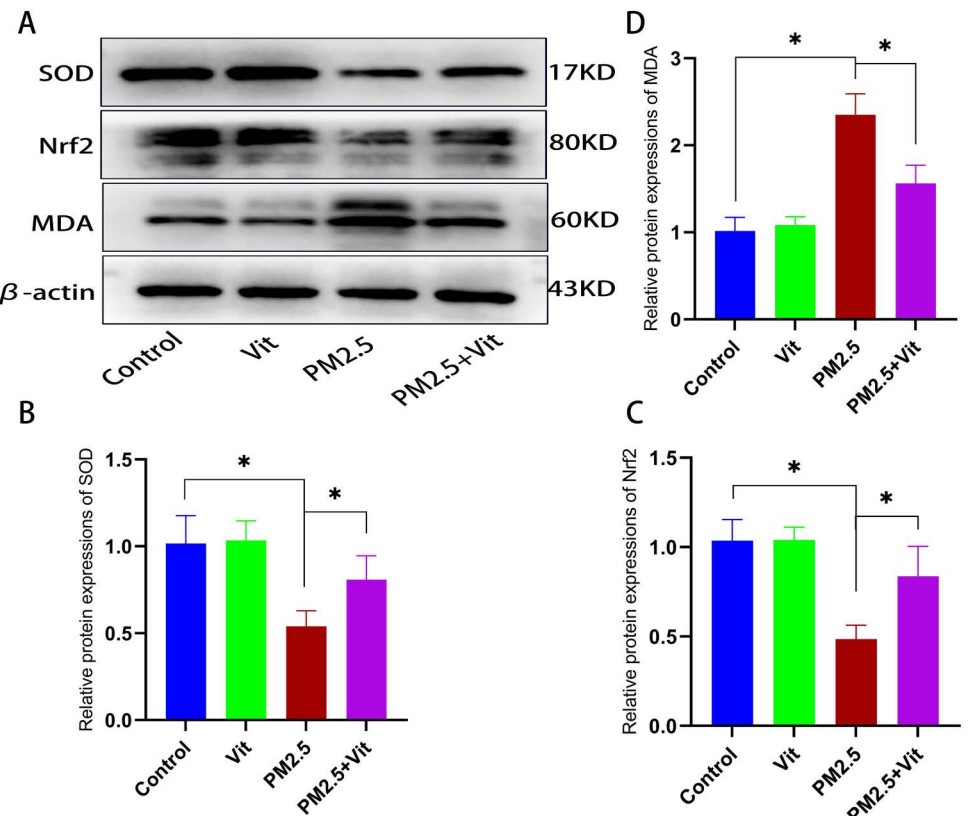

**Fig 2. Oxidative stress (OS) responses in the testis tissues.** A) Bands of OS-related proteins in the testis tissue. B-D) Quantification of protein expressions. SOD and Nrf2 expressions were downregulated, and MDA expression was upregulated after PM2.5 exposure, which was reversed after vitamin intervention. The assay was repeated thrice, and the data are presented as means ± standard deviation. * $p < 0.05$ *vs.* control rats, #$p < 0.05$ *vs*. PM2.5 rats. https://doi.org/10.5061/dryad.8kprr4xxq.

mitochondria were oval, evenly distributed, abundant, and had visible and uniform cristae. After PM2.5 exposure, the number of mitochondria decreased, mitochondrial swelling occurred, and the mitochondrial cristae were disordered and even disappeared with vacuolation, which all improved to a certain extent after vitamin interference. In addition, WB was carried out to determine expressions of core UPR^mt proteins (Fig 4 and S2 File). The protein expressions of HSP60, CHOP, and ATF5 significantly decreased in the PM2.5 versus control rats ($p < 0.05$), and they increased in the PM2.5 + Vit versus PM2.5 rats ($p < 0.05$).

### 3.7 Apoptosis of testis tissues

As shown in Fig 5 and S2 File, WB revealed a marked decline of anti-apoptotic protein B-cell lymphoma-2 (Bcl-2), alongside elevations in the pro-apoptotic proteins Bcl-2 associated X protein (Bax) and Caspase3 in the PM2.5 versus control rats ($p < 0.05$). Relative to the PM2.5 rats, the PM2.5 + Vit rats had significantly downregulated expressions of Bax and Caspase3 and an upregulated Bcl-2 content ($p < 0.05$). Moreover, using TUNEL assay, we quantified testis tissue apoptosis (Fig 6 and S3 File). The apoptosis index was substantially elevated among PM2.5 versus control rats ($p < 0.05$), whereas it was drastically decreased among PM2.5 + Vit rats ($p < 0.05$).

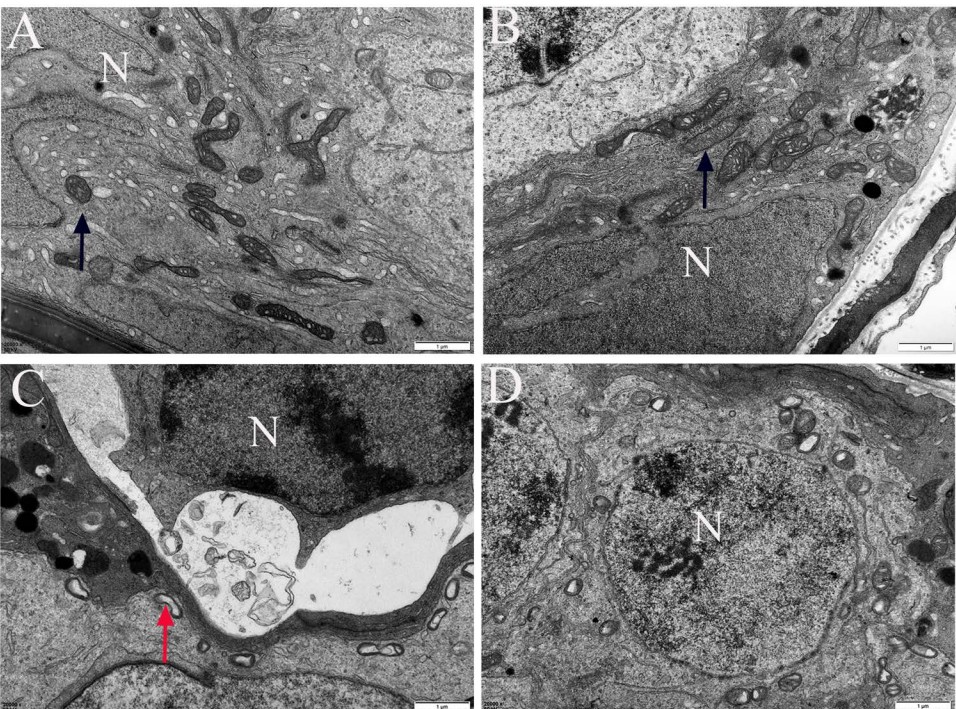

**Fig 3. Ultrastructure of mitochondria in germ cells.** A) In controls, mitochondria were distributed evenly, and the mitochondrial cristae were visible and intact. B) In the Vit group, the mitochondrial structure was intact. C) In PM2.5 rats, mitochondria were distributed sparsely, mitochondrial swelling occurred, and the mitochondrial cristae were unclear and even disappeared. D) In PM2.5 + Vit rats, the mitochondrial morphology was improved. **N:** Nucleus. The black arrow indicates normal mitochondrial morphology, and the red arrow indicates mitochondrial vacuolation; scale bar = 1 μm. https://doi.org/10.5061/dryad.pzgmsbcz2.

## 4. Discussion

Our conclusions present novel insights into the molecular response of male reproductive health to PM2.5 exposure. High concentrations of PM2.5 induce SOD and Nrf2 antioxidant responses together with mitochondrial stress and interfere with UPR^mt, mitochondrial function, and apoptosis, creating a new expression mechanism.

Automobile exhaust has become the primary source of atmospheric pollution in large and medium-sized cities, where PM2.5 has the most complex composition and the strongest pathogenicity[12]. Studies have shown that PM2.5 carries a variety of toxic heavy metals and polycyclic aromatic hydrocarbons (PAHs), which enter the human body via breathing to cause functional damage to multiple organs and tissues [13]. The damage from PM2.5 to the reproductive system has been of concern to countries worldwide. Research suggests that PM2.5 can lead to sperm quality decline, causing male infertility [14, 15]. Nonetheless, the associated signaling network is currently unknown.

Herein, 4-week-old rats were selected, and 10 mg/kg was used as the PM2.5 exposure dose according to the current Ambient Air Quality Standards (GB3095-2012) and the previous experimental results of our group [16]. The strong antioxidant activity of vitamin C plus vitamin E has been verified [16], and the dose and method used in this study have been confirmed to have no toxicological effect on animals. After four weeks of PM2.5 administration, the body, testis and epididymis weights decreased, the morphological

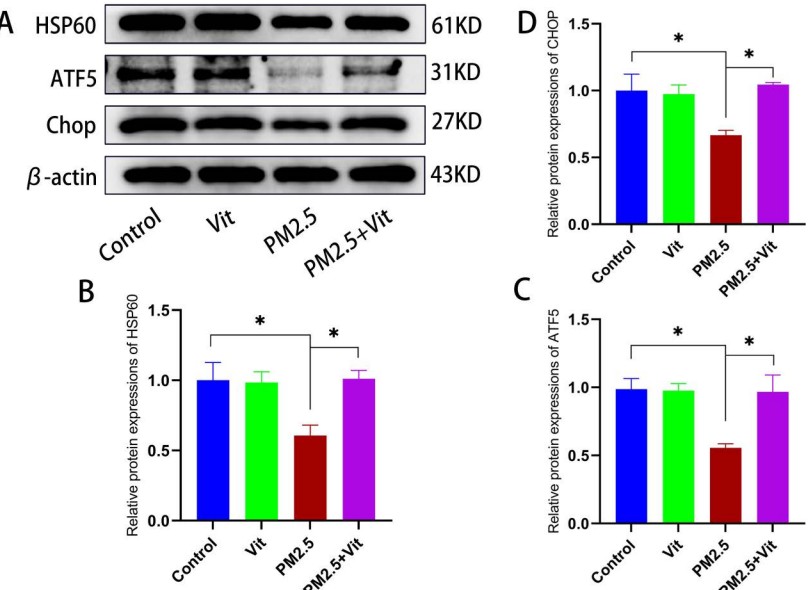

**Fig 4. Expressions of core UPRmt proteins following PM2.5 exposure.** A) WB bands of marker UPRmt proteins. B) Quantitative analysis of protein expressions. PM2.5 exposure drastically diminished HSP60, ATF5, and CHOP contents, indicating UPRmt impairment. After vitamin intervention, UPRmt was activated, and related protein expressions were upregulated. * $p < 0.05$ *vs.* control rats, #$p < 0.05$ *vs.* PM2.5 rats. https://doi.org/10.5061/dryad.8kprr4xxq.

structure of the testis changed, the seminiferous tubules were poorly developed, the quantity of spermatogenic cells diminished, the apoptosis index of testis tissues increased, both sperm count and motility decreased, and the sperm abnormality rate increased. The above results show that PM2.5 from automobile exhaust significantly impair male reproductive functions. After combined vitamin intervention, the body and organ weights of the testis and epididymis increased, the damage to testis tissues was relieved, sperm count and motility increased, and the sperm abnormality rate decreased. Thus, combined vitamin intervention can alleviate the male reproductive impairment brought on by PM2.5 from automobile exhaust.

The OS response is a primary cellular stress response. Under endogenous or exogenous stimuli, tissues and cells will generate excess ROS, inducing OS [17]. Mitochondrial stress contributes to ROS generation [18]. OS is potentially a major contributor to PM2.5-driven germ cell damage. The potential of PM2.5 to generate ROS in multiple cell types has been verified, which is linked to lung injury, nerve damage, and endocrine disturbances [19]. Excessive or sustained OS can cause apoptosis[20]. Based on these theories, we conducted WB analysis. We revealed that the SOD and Nrf2 contents were diminished, but the MDA content was increased in the testis tissue of PM2.5 rats. Alterations in these crucial parameters for anti-OS responses indicate that PM2.5 from automobile exhaust can induce ROS generation and subsequent OS responses in testis tissue. However, combined vitamin intervention alleviates OS responses and effectively prevents germ cell damage.

Mitochondria are the primary sites of ROS production and processing. Excess ROS augments mitochondrial membrane permeability, mitochondrial swelling, mitochondrial permeability transition pore (MPTP) opening, and mitochondrial DNA (mtDNA) damage [21]. Decreases in ROS activity can reduce ROS-related mitochondrial damage, maintaining mitochondrial function [22]. Moreover, ROS have various signaling functions, so

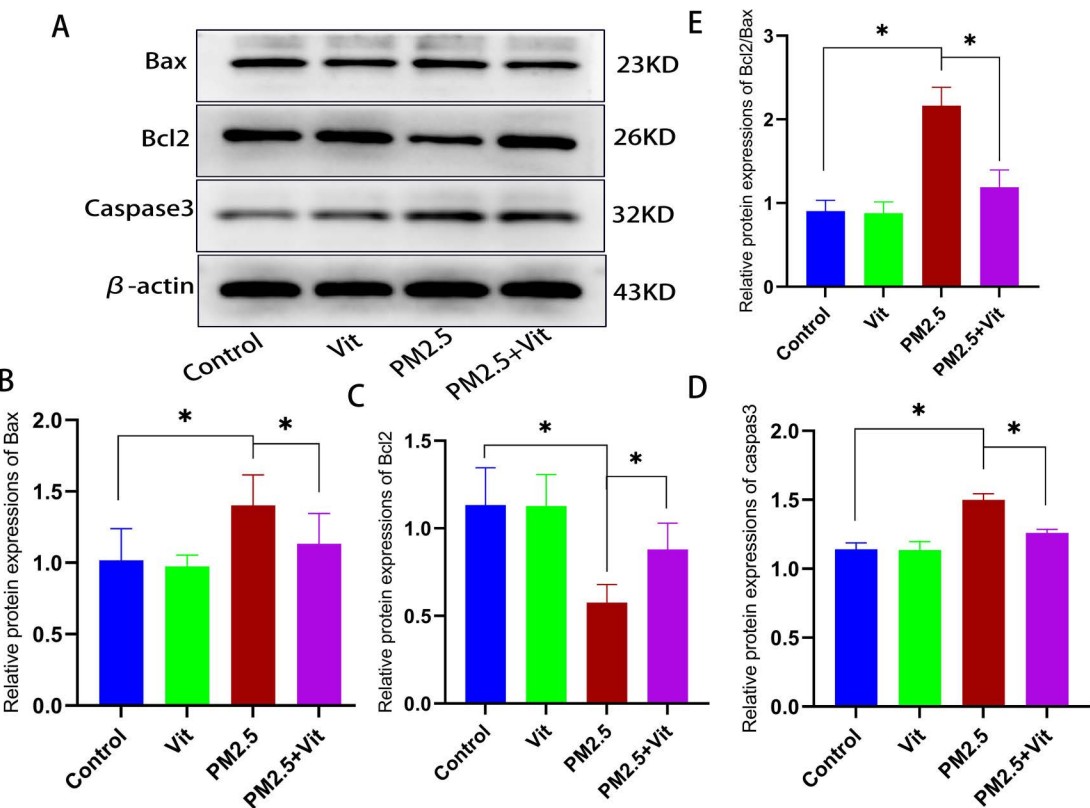

**Fig 5. Testis tissue apoptosis post PM2.5 exposure.** A) WB of Bax, Bcl-2, and Caspase3. B) Quantification of protein expressions. After PM2.5 exposure, Bax and Caspase3 expressions were significantly enhanced, whereas Bcl-2 was diminished, suggesting that PM2.5 exposure enhances apoptosis. Apoptosis was weakened after vitamin intervention. * $p < 0.05$ *vs*. control rats, # $p < 0.05$ *vs*. PM2.5 rats. https://doi.org/10.5061/dryad.8kprr4xxq.

ROS produced by mitochondria can serve as signals of mitochondrial functional status [23]. Mitochondrial damage leads to misfolding or unfolding of proteins in mitochondria so that excess ROS will be produced to activate UPRmt [24]. Then, activated UPRmt helps restore protein folding and degrades damaged proteins to maintain protein balance in cells. This can maintain mitochondrial protein homeostasis, restore bioactive functions, and help achieve cell survival. In this study, PM2.5 triggered OS responses in germ cells, and a large amount of ROS contributed to mitochondrial swelling and structural changes, as well as reduced expressions of HSP60, CHOP, and ATF5, demonstrating that PM2.5 can also induce UPRmt in germ cells.

UPRmt is involved in multiple diseases that share a common pathogenesis of mitochondrial dysfunction. Short-term or mild UPRmt is beneficial to the body because it can overcome initial damage, whereas long-term or high-intensity activation of UPRmt is harmful, possibly because UPRmt is impaired and finally weakened due to depletion of UPRmt-related proteins, losing its protection of mitochondria and promoting apoptosis [25, 26]. In this study, after high PM2.5 administration, the SOD and Nrf2 contents were drastically diminished, whereas MDA expression significantly enhanced. Interestingly, the essential UPRmt proteins (CHOP, HSP60, and ATF5) were significantly downregulated, accompanied by mitochondria swelling, ultrastructural disorders, and significant increases pro-apoptotic proteins Bax and Caspase3 expressions, along with TUNEL-positive cells. Thus, PM2.5 exposure

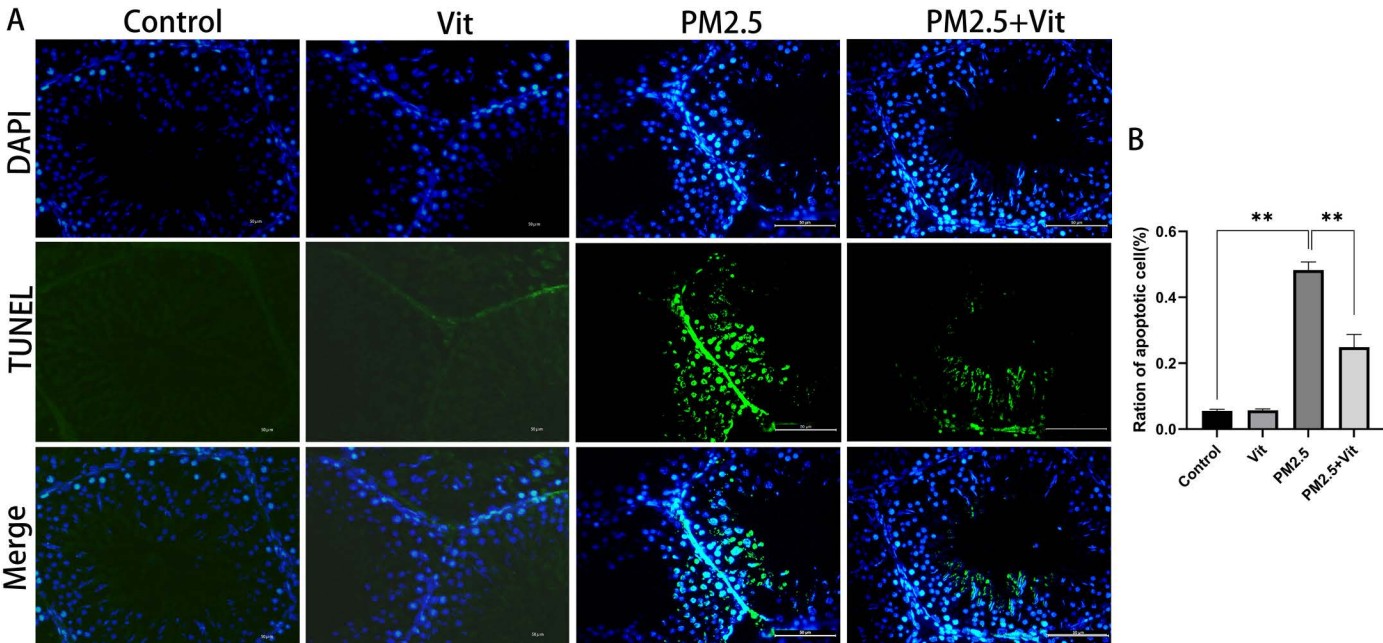

**Fig 6. TUNEL staining of testis tissues.** A) Apoptosis observed using a fluorescence microscope. Apoptosis increased following PM2.5 exposure and decreased after vitamin administration. B) Apoptosis index. The data shown are the means ± standard deviation. **$p < 0.01$. Scale bar = 50 μm. https://doi.org/10.5061/dryad.pzgmsbcz2.

at 10 mg/kg from automobile exhaust can cause massive production of ROS in cells and break the oxidation/antioxidation balance, ultimately depleting UPRmt, followed by aggravation of mitochondrial damage, cell dyshomeostasis, activation of the apoptosis pathway, and apoptosis. Meanwhile, combined vitamin intervention was found to relieve OS responses. It reactivated UPRmt by appropriate amounts of ROS, thus protecting cell homeostasis and inhibiting pro-apoptotic protein expressions and apoptosis of testis tissues. This mirrors a prior analysis whereby inhibiting ROS generation was found to protect mitochondria via UPRmt, thus protecting cardiomyocytes from ischemia/reperfusion injury [27]. Therefore, ROS plays a vital role in UPRmt, which can both activate and eventually weaken UPRmt, and combined vitamins regulate UPRmt by inhibiting ROS. Therefore, PM2.5 from automobile exhaust may mediate reproductive dysfunction in male rats via the ROS and UPRmt pathways, and combined vitamins may protect against automobile exhaust-derived PM2.5-mediated reproductive impairment.

Overall, this study showed that automobile exhaust-derived PM2.5 might induce reproductive impairment in male rats via the ROS-UPRmt pathway, and vitamin C plus vitamin E at appropriate doses protected germ cells by regulating ROS-UPRmt. However, these data were derived from *in vivo* experiments, and the biological function of UPRmt is complex. Thus, the exact mechanism needs to be further studied.

## 5. Conclusion

PM2.5 from automobile exhaust causes male reproductive impairment, and the possible mechanism is as follows. PM2.5 damages mitochondria through the ROS-UPRmt pathway, resulting in germ cell apoptosis. Combined vitamins can reduce ROS production to improve UPRmt function and protect mitochondria, thus reducing apoptosis. This study helps us

understand better the mechanism of male infertility caused by atmospheric pollution, proz-vides new insights into PM2.5-mediated reproductive toxicity, and identifies new therapeutic targets for male infertility.

## Supporting information

**S1 File. Animal phenotype raw data.** Body weight, litter size, epididymal sperm count and Organ Index of male SD rats are provided as original data.
(XLSX)

**S2 File. Raw images of gels and blots.**
(ZIP)

**S3 File. Original images of HE staining, transmission electron microscopy, and fluorescence.**
(ZIP)

## Acknowledgments

The authors would like to extend their gratitude to Dr. Bin Liu for his technical assistance. We are also deeply appreciative of the expertise provided by Researcher Guanghai Liu in the field of testicular morphology. Furthermore, we thank the Advanced Optical Microscopy Technical Platform at the Clinical Experimental Center of Zunyi Medical University for their invaluable support.

## Author contributions

**Conceptualization:** Cao Wang, Bin Liu.

**Data curation:** Yingchi Zhao, Zhen Luo, Kaiyi Mao.

**Formal analysis:** Yingchi Zhao, Zhen Luo.

**Funding acquisition:** Cao Wang.

**Methodology:** Yingchi Zhao, Zhen Luo, Kaiyi Mao.

**Project administration:** Cao Wang.

**Resources:** Bin Liu, Guangxu Zhou.

**Software:** Yingchi Zhao, Zhen Luo, Guangxu Zhou, Kaiyi Mao.

**Supervision:** Cao Wang, Bin Liu.

**Visualization:** Guangxu Zhou.

**Writing – original draft:** Cao Wang.

**Writing – review & editing:** Cao Wang.

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
