## [Decision Letter · Decision Letter 0]

1 Dec 2024

PONE-D-24-49516PM2.5 from Automobile Exhaust Induces Apoptosis in Male Rat Germ Cells via the ROS-UPRmt Signaling PathwayPLOS ONE

Dear Dr. Wang,

Thank you for submitting your manuscript to PLOS ONE. After careful consideration, we feel that it has merit but does not fully meet PLOS ONE’s publication criteria as it currently stands. Therefore, we invite you to submit a revised version of the manuscript that addresses the points raised during the review process.

We look forward to receiving your revised manuscript.

Kind regards,

Ming Zhang, Ph.D.

Academic Editor

PLOS ONE

5. We note that your Data Availability Statement is currently as follows: All relevant data are within the manuscript and its Supporting Information files.

Reviewers' comments:

Reviewer's Responses to Questions

**Comments to the Author**

1. Is the manuscript technically sound, and do the data support the conclusions?

Reviewer #1: Yes

Reviewer #2: Partly

Reviewer #3: Yes

2. Has the statistical analysis been performed appropriately and rigorously? 

Reviewer #1: Yes

Reviewer #2: No

Reviewer #3: Yes

3. Have the authors made all data underlying the findings in their manuscript fully available?

Reviewer #1: Yes

Reviewer #2: Yes

Reviewer #3: Yes

4. Is the manuscript presented in an intelligible fashion and written in standard English?

Reviewer #1: Yes

Reviewer #2: Yes

Reviewer #3: Yes

5. Review Comments to the Author

Reviewer #1: 1, methods said "PM2.5 was collected using atmospheric particulate samplers on heavily traveled roads", but this cannot ensured that the PM2.5 is from vehicle exhaust. Author should show the difference between the PM2.5 used in this manuscript and standard PM2.5. Otherwise, it is not proper to emphasize that PM2.5 is from vehicle exhaust. Since this is cirtical for the manuscript, author should provide the evidence that PM2.5 from vehicle exhaust is special or different from standard PM2.5.

2，In line 280, author wrote "High concentrations of PM2.5 induce SOD and Nrf2 antioxidant....". but in this manuscript, there are no comparisons between the low dosage and high dosage PM2.5, therefore this statement is wrong.

3, PM2.5 can not only induce apoptosis, but also pyroptosis (PMID: 37236293). it worths an investigation on the pyroptosis as well.

minor points

1, the term, like "UPRmt", should mt be the superscript？please keep it consistent in the manuscript.

2, the catalog number of all the antibodies and chemicals should be provided.

Reviewer #2: The manuscript written by Cao Wang et al. is to explore how fine particulate matter (PM2.5) from vehicle exhaust affects reproductive function in male rats and to determine the role of vitamins in this process. There are some comments as follows.

1.It is better to add some information of vitamin role in the reproduction.

2.Please provide the reason of 100 mg/kg vitamin C and 50 mg/kg vitamin E intervention at the same time, not individually.

3.A 4-week exposure was chosen in this study. What is the reason?

4.In view of the authenticity of the results, it is recommended to re-analyze the experimental results.

5.In lines 312-313 of the Discussion section, the authors conclude that “combined vitamin intervention alleviates OS responses and effectively prevents or cures germ cell damage.” No such conclusion can be drawn from the results of the experiment, so it is suggested to revise this sentence.

6.Please check the References carefully and modify them as requested by the Journal.

Reviewer #3: The manuscript investigated the influence of PM2.5 particles on the reproductive capacity of male rats. Overall, the study appears to be well-designed, and the writing is coherent. However, a few minor points for consideration are as follows:

(1) The study administered PM2.5 particles via transnasal drip of the collected particles. How representative is this method of the typical human exposure route, which is through inhaling contaminated air? It would be beneficial to discuss the implications of this administration method for translating the findings to human health effects.

(2) There lacks statistical analysis to compare the differences between groups presented in Table 2.

(3) The manuscript lacks summary statistics and a comparison of cell numbers under different conditions in Figure 1.

6. PLOS authors have the option to publish the peer review history of their article (what does this mean? ). If published, this will include your full peer review and any attached files.

**Do you want your identity to be public for this peer review?** For information about this choice, including consent withdrawal, please see our Privacy Policy .

Reviewer #1: No

Reviewer #2: No

Reviewer #3: No

---

## [Author Response · Author response to Decision Letter 1]

1 Jan 2025

Dear Editor,

Thank you very much for giving us an opportunity to revise our manuscript entitled “PM2.5 from Automobile Exhaust Induces Apoptosis in Male Rat Germ Cells via the ROS-UPRmt Signaling Pathway”. We also thank the reviewers for the constructive comments and suggestions. We have revised the manuscript accordingly, and all amendments are indicated by red font in the revised manuscript. In addition, our responses to these comments are listed below this letter.

We hope that our revised manuscript is now acceptable for publication in your journal and look forward to hearing from you soon.

With best wishes,

Yours sincerely,

Cao Wang

Replies to Reviewer 1

1, methods said "PM2.5 was collected using atmospheric particulate samplers on heavily traveled roads", but this cannot ensured that the PM2.5 is from vehicle exhaust. Author should show the difference between the PM2.5 used in this manuscript and standard PM2.5. Otherwise, it is not proper to emphasize that PM2.5 is from vehicle exhaust. Since this is cirtical for the manuscript, author should provide the evidence that PM2.5 from vehicle exhaust is special or different from standard PM2.5.

Response: Thank you for your suggestions. The particles collected from busy urban traffic arteries indeed cannot guarantee 100% origin from vehicle exhaust. However, it is certain that vehicle exhaust emissions constitute the primary component. In our preliminary experiments, we purchased a standard PM2.5 sample derived from U.S. vehicle exhaust (Catalog No.: 1648a) for comparative studies, and both caused no differences in reproductive damage to male SD rats. In future studies, we will conduct a component analysis of PM2.5 derived from vehicle exhaust to identify the primary substances harmful to the reproductive system.

2，In line 280, author wrote "High concentrations of PM2.5 induce SOD and Nrf2 antioxidant....". but in this manuscript, there are no comparisons between the low dosage and high dosage PM2.5, therefore this statement is wrong.

Response: Thank you for your valuable suggestions，Corrections have been made in the revised manuscript.

3, PM2.5 can not only induce apoptosis, but also pyroptosis (PMID: 37236293). it worths an investigation on the pyroptosis as well.

Response: Thank you for your valuable suggestions, which will guide our next steps in the research.

4,minor points

(1), the term, like "UPRmt", should mt be the superscript？please keep it consistent in the manuscript.

Response:Thank you for your suggestion. Yes, “mt” in UPRmt requires a superscript, which has been corrected in the revised manuscript to ensure consistency throughout the document.

(2), the catalog number of all the antibodies and chemicals should be provided.

Response: Thank you for your advice. We have added numbering for antibodies and chemicals in the manuscript.（Page 8,lines146-148）.

Replies to Reviewer 2

1.It is better to add some information of vitamin role in the reproduction.

Response: Thank you for your valuable suggestions， We have added sentences and marked them in red in the revised manuscript（Page4,Lines73-75）.

2.Please provide the reason of 100 mg/kg vitamin C and 50 mg/kg vitamin E intervention at the same time, not individually.

Response: We have already confirmed the effect of combined vitamin use on male reproduction at this dose in our previous studies. (PMID: 30326357).

3.A 4-week exposure was chosen in this study. What is the reason?

Response: Thank you for your suggestion. We have conducted preliminary experiments and successfully established an animal exposure model with satisfactory results, which is why we have been consistently using it up to today. (PMID: 30326357).

4.In view of the authenticity of the results, it is recommended to re-analyze the experimental results.

Response: Thank you for your suggestion. We have reorganized and analyzed the experimental data.

5.In lines 312-313 of the Discussion section, the authors conclude that “combined vitamin intervention alleviates OS responses and effectively prevents or cures germ cell damage.” No such conclusion can be drawn from the results of the experiment, so it is suggested to revise this sentence.

Response: Thank you for your suggestions. We have revised the manuscript according to your valuable feedback（Page 18,lines312）.

6.Please check the References carefully and modify them as requested by the Journal.

Response: Thank you for your advice. The format of references in the revised manuscript has been corrected as required by the journal.

Replies to Reviewer 3

1. The study administered PM2.5 particles via transnasal drip of the collected particles. How representative is this method of the typical human exposure route, which is through inhaling contaminated air? It would be beneficial to discuss the implications of this administration method for translating the findings to human health effects.

Response: You’re welcome for the suggestion. During our initial experiments, we anesthetized Sprague-Dawley rats and administered the drug via tracheal intubation. This method, however, caused trauma, potentially damaging the trachea and increasing asphyxiation risk. Through literature review, we discovered nasal instillation, which we tested in pilot studies. It produced similar results to tracheal administration but better simulated human conditions and reduced anesthesia and intubation risks.

2. There lacks statistical analysis to compare the differences between groups presented in Table 2.

Response: Thank you for your suggestions. We have revised the manuscript according to your valuable feedback（Page 9,lines173-174）.

3. The manuscript lacks summary statistics and a comparison of cell numbers under different conditions in Figure 1.

Response: Thank you for your insightful suggestion. We are here to show the morphological changes of testicular tissue, and our quantitative analysis is demonstrated by WB detection of apoptotic proteins Bax and Bcl-2 and TUNEL staining of testicular tissue.

---

## [Editor Report · Decision Letter 1]

10 Feb 2025

PM2.5 from Automobile Exhaust Induces Apoptosis in Male Rat Germ Cells via the ROS-UPRmt Signaling Pathway

PONE-D-24-49516R1

Dear Dr. Wang,

We’re pleased to inform you that your manuscript has been judged scientifically suitable for publication and will be formally accepted for publication once it meets all outstanding technical requirements.

Kind regards,

Ming Zhang, Ph.D.

Academic Editor

PLOS ONE

---

## [Editor Report · Acceptance letter]

PONE-D-24-49516R1

PLOS ONE

Dear Dr. Wang,

I'm pleased to inform you that your manuscript has been deemed suitable for publication in PLOS ONE. Congratulations! Your manuscript is now being handed over to our production team.

Kind regards,

on behalf of

Dr. Ming Zhang

Academic Editor

PLOS ONE